# Study on Dynamic Response Characteristics of Circular Extended Foundation of Large Wind Turbine Generator

**Zong-Wei Deng** [1] [ID], **Zi-Jian Fan** [2],*[ID], **Yan-Ming Zhou** [2] **and Pei-Yu Deng** [2]

1   College of Civil Engineering, Hunan City University, Yiyang 413000, China
2   School of Civil Engineering, Changsha University of Science & Technology, Changsha 410114, China
*   Correspondence: fzj_sophia@stu.usc.edu.cn

**Abstract:** In order to study the dynamic response characteristics of circular extended foundation of wind turbine in mountainous areas, a 1:10 scaled model test was carried out on the circular extended foundation of 2MW wind turbine, and the deformation characteristics of wind turbine foundation under random wind load were analyzed by ABAQUS numerical calculation. The results show that: (1) The wind turbine foundation has different stress types on the windward side and the leeward side. The components of the windward side foundation are subjected to tensile stress, while the components of the wind turbine leeward side foundation are subjected to compressive stress. (2) The strain of the foundation bolt, the strain of the foundation ring, and the strain of the foundation plate are within the allowable range of material deformation, but the relative deformation of the windward side and the leeward side is quite different. (3) The numerical calculation results of wind turbine foundation under strong wind load are compared with the failure results of scale model experiment, which shows that the overall overturning failure of foundation is a dynamic response mode of wind turbine foundation. In the design and construction, it is necessary to strengthen the research on the windward side and the leeward side and strengthen the anti-overturning design of the wind turbine expansion foundation.

**Keywords:** wind turbine; dynamic response; wind loading; spread footing

## 1. Introduction

Wind turbines are power generation facilities built in wind-rich areas where wind loads are highly variable. The direction and intensity of the wind always fluctuate with time, which makes it difficult to simulate the force of each component of the wind turbine. The change of the wind angle from 30 degrees to 40 degrees will cause the thrust of the wind turbine to change by 27% [1–3], and the wind is also different at different heights [4–7]. As a complex structure, the wind turbine structure is easily damaged during the working process. As a transfer structure of the force load, its foundation not only bears a large vertical load, but also bears a large horizontal force and overturning moment [8]. Due to this characteristic, the wind turbine foundation and the components connected to the tower need to be studied.

The dynamic response characteristics of wind turbine structures have been extensively studied, mainly focusing on different components of wind turbines [9–12]. However, domestic and foreign scholars on the basic characteristics of wind turbine damage have mainly concentrated in the upper part of the wind turbine structure. Aiming at the failure characteristics of wind turbine foundation under bending moment load [13–16], Xu et al. [17] studied the dynamic response characteristics of three different forms of offshore wind turbine pile foundations under seismic load and obtained the dynamic response characteristics of different pile foundation types under seismic load. Wang et al. [18] analyzed the load characteristics of single pile foundation of wind turbine in soft soil multilayer foundation and obtained the development and failure mode of single pile foundation under

different loads. Wu et al. [19] studied the horizontal force characteristics of single pile foundation of offshore wind turbine under soft soil conditions and obtained the maximum bending moment of pile body and the maximum pullout force of soil around pile. Wu [20] studied the failure model of offshore single pile foundation under horizontal cyclic loading by using a new horizontal loading device. Lu et al. [21] and He et al. [22] analyzed the fatigue damage effect of foundation ring and concrete joint based on engineering examples. Stamatopoulos [23] studied the seismic performance of wind turbines supported by circular extended foundations using current Greek seismic codes. Dai [24–28] evaluated that the structural response of the embedded ring showed significant vertical displacement during turbulent wind speed.

At present, the research of onshore wind turbine foundation is far less than that of offshore wind turbine foundation, but in different working conditions, the criterion of foundation damage is the same [29–32]. In the mountain wind turbine foundation type, the most widely used is the expansion foundation [4]. Kozmar [33] carried out wind tunnel experiments on wind turbines parked in different mountainous environments and found that the gas flow delay of wind turbines was more obvious near the surface and mountains, and the disturbance of tangential airflow was closely related to the mountainous environment. Aiming at the shortcomings of the existing mountain wind turbine foundation, Li et al. [34] proposed a new type of conical wind turbine foundation model and analyzed the force characteristics of the model. Michel [35] studied the dynamic response characteristics of wind turbine foundation in seismic active areas and soft soil areas. It is found that the type of soil has a great influence on the dynamic response of the foundation, and the natural frequency of the wind turbine foundation will increase on some soil foundations. Mohamed [36] proposed a honeycomb raft foundation with an active stabilization system and compared it with the bearing capacity of the existing foundation. It is found that the foundation can significantly improve the basic capacity and can be reused on an existing basis.

In the mountainous environment, the circular expansion foundation is still the main foundation form adopted by the wind turbine. In the past, the research work of mountain wind turbine mainly focused on the force characteristics of its superstructure and the wake characteristics of wind turbine structure. Under the action of random wind load, the research on the mechanical characteristics of the components of the wind turbine foundation structure and the dynamic response characteristics of the foundation has not attracted attention. In this paper, the circular expansion foundation of 2MW installed capacity is taken as the research object, a 1:10 scale model is established, and ABAQUS is used to calculate the wind turbine foundation. The connecting bolts, foundation ring, bottom plate strain, foundation displacement and foundation pressure of the foundation and tower are studied, and the dynamic response change modes of the extended foundation model experiment and numerical calculation of the wind turbine are compared and analyzed. A targeted solution is proposed according to the research results in this paper.

## 2. Materials

### 2.1. Test Material

Before establishing the model experiment, based on the similarity theory and dimensional analysis method, the geometric similarity ratio was selected as the similarity constant, and the material similarity constant and load similarity constant of the model were obtained according to the conversion formula. In the similarity ratio calculation, the influence of the small components of the structure is ignored, and the influence of the upper wind load on the foundation is considered as a whole. The material parameters of the wind turbine foundation are shown in Table 1.

**Table 1.** Model material mechanics parameters.

| Position | Material | Parameter | Numerical Value |
|---|---|---|---|
| Tower | Steel | E | 212.8 GPa |
| | | $\mu$ | 0.31 |
| | | $\sigma_s$ | 349.2 MPa |
| | | $\rho$ | 7.85 g·cm$^{-3}$ |
| Reinforced concrete foundation | Concrete | E | 32.3 MPa |
| | | $\mu$ | 0.23 |
| | | $f_{cu.k}$ | 27.7 MPa |
| | | $\rho$ | 0.0234 g·m$^{-3}$ |
| | Reinforcement | E | 203.8 MPa |
| | | $\mu$ | 0.31 |
| | | $\sigma_s$ | 237.6 MPa |
| | | $\rho$ | 7.85 g·cm$^{-3}$ |
| Foundation soil | Undisturbed soil | $E_s$ | 8.35 MPa |
| | | $\mu$ | 0.33 |
| | | $\psi$ | 23.2 |
| | | $\rho$ | 2.03 g·cm$^{-3}$ |
| | | c | 17.6 kPa |
| | | $\omega$ | 16.7% |

In this test, the model filling is made of 2 mm sieved residual soil and a certain amount of sand. The gradation curve of the soil used in the test is shown in Figure 1, and its non-uniformity coefficient is 10.2. Therefore, it is necessary to add sand to improve the gradation of the soil to meet the conversion requirements for controlling the shear strength of the model foundation. In addition to optimizing the soil gradation, the foundation soil of the model is compacted manually to ensure the soil compactness requirements.

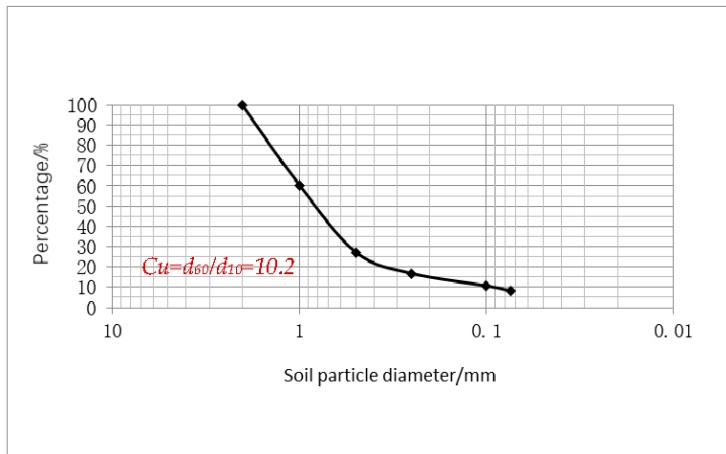

**Figure 1.** Particle grading curve.

The test mainly explores the dynamic response characteristics of the wind turbine foundation. In the indoor test, it cannot fully meet the requirements of the scale test. In order to effectively explore its response characteristics, the experimental conditions are simplified, and the smaller part of the correlation with the research object is transformed into other factors. The tower model only scales the bottom section of the wind turbine tower prototype, while the components of the wind turbine nacelle and blades are expressed by load. It mainly tests the bolt components, bottom tower, and foundation ring connected to the tower model and foundation. Considering that the wall thickness of the scaled tower model is only 3 mm, in order to ensure the safety of the test and ensure the similarity of its parameters, the tower model is made of the same material as the prototype tower, and the tower strain data obtained by calculations were analyzed. The reinforced concrete

foundation is simulated by simulated concrete, and the maximum aggregate size is selected as 1/10 of the actual maximum aggregate size. The cushion concrete is simulated by cement mortar, and the steel wire is used to ensure the reinforcement ratio.

### 2.2. Simplification of Load

The hydrodynamic calculation and theoretical analysis of the horizontal thrust of the wind turbine and the wind load of the tower are carried out. It is found that the wind load acting on the tower accounts for about 15.6% of the horizontal wind load of the whole wind turbine [37], and the variation of the wind load of the tower cannot be ignored. In order to simplify the loading conditions of the experimental load, the random wind load of the wind turbine is calculated, and the load value loaded on the model is controlled according to the load similarity constant and the bending moment of the tower. Therefore, in the follow-up study of this paper, Fr is considered as the dynamic load acting on the wind turbine, and Fz, Mr, G1, and G2 are considered as the variable load acting on the wind turbine foundation.

The analysis of the components of the wind turbine shows that the weight of the wind turbine, generator, engine room, and tower constitute the vertical force $F_z$; the resultant force of wind wheel horizontal thrust $F_{XH}$ and tower wind load $P_z$ composes horizontal resultant force Fr; the resultant moment of the wind wheel torque $M_{XH}$, the wind wheel pitch moment $M_Y$, and the additional moment $M_{YT}$ caused by the mass eccentricity constitute the horizontal resultant moment Mr. Among the above load components, the wind wheel horizontal thrust $F_{XH}$, the tower wind load $P_z$, the wind wheel torque $M_{XH}$, and the wind wheel pitch moment $M_Y$ are greatly affected by random wind.

The analysis of the force of the wind turbine shows that the wind turbine foundation is mainly affected by permanent load and variable load. The permanent load is composed of foundation gravity, superstructure vertical load, and backfill gravity. The horizontal loads in the x and y directions and the moments in the x, y, and z directions caused by the wind load are composed of variable loads, which are represented by Fx, Fy, Mx, My, and Mz. The six variable loads on the wind turbine are represented in the basic coordinate system of the wind turbine, as shown in Figure 2. According to the load calculation method recommended by China's wind turbine foundation design, the force on the same horizontal plane can be combined into a horizontal force, and the moment on the same horizontal plane can be combined into a resultant moment. Since the wind turbine is mainly affected by the horizontal random wind load, the value of Mz is one order of magnitude smaller than Mr, and the moment Mz caused by the vertical additional load of the wind turbine is usually not considered. Therefore, the variable load is finally simplified to vertical load Fz, horizontal force Fr, and horizontal moment Mr.

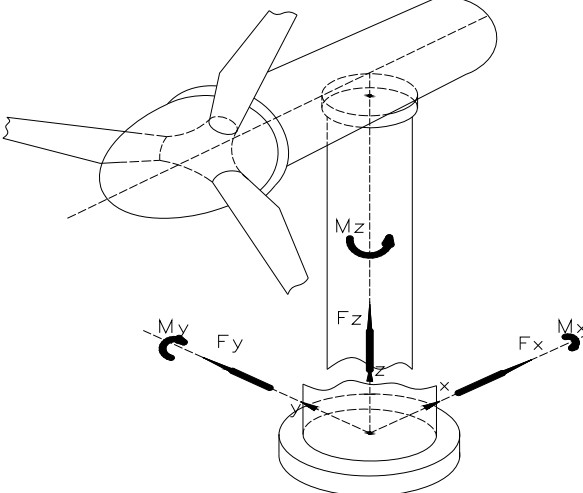

**Figure 2.** The load acting on the foundation.

In order to accurately simulate the wind load conditions in the natural state and highlight the additional influence of wind load on the foundation, it is assumed that the wind turbine is mainly affected by the horizontal fluctuating wind load, and its load curve is converted by the Davenport wind speed spectrum. The wind-speed spectrum considers the random fluctuation law of the average wind and periodic variation of the wind speed. The wind load operation period is 10 min, and the variation range is 0.2 Fr. By using MATLAB software for random calculation, the wind curve in this paper is shown in Figure 3.

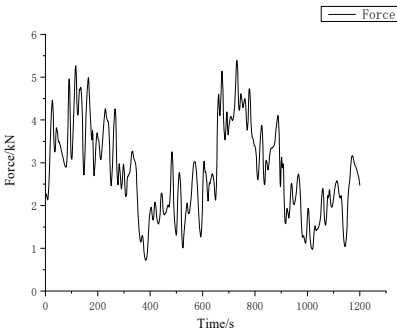

**Figure 3.** The level of random wind load.

### 2.3. Loading Device and Observation Instrument Arrangement

During the test, the force in the horizontal direction of the wind turbine was provided by the horizontal servo actuator, the vertical load was applied by a jack perpendicular to the tower, and the torque was applied by an additional load with a certain eccentricity. The specific installation of the model is shown in Figure 4.

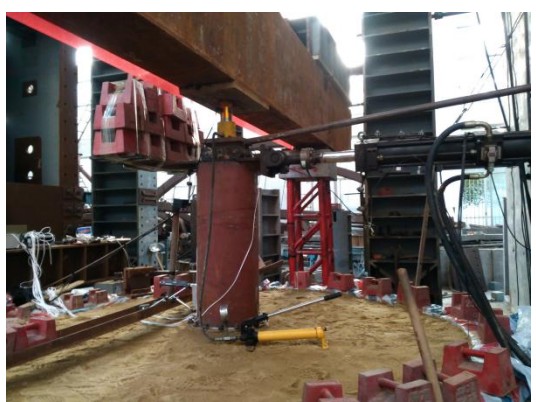

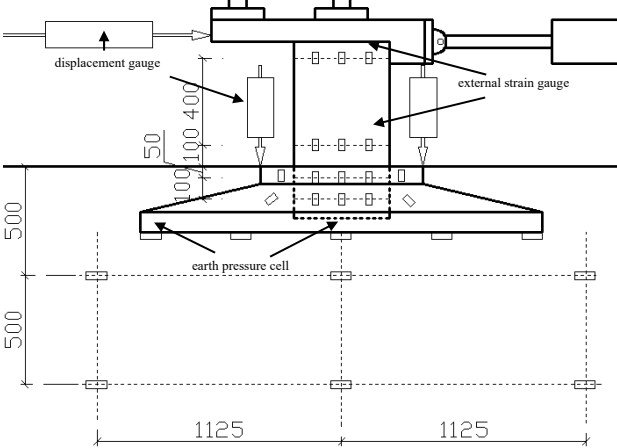

**Figure 4.** Loading device arrangement.

Horizontal displacement meters and vertical displacement meters were arranged on the top of the foundation to test the change of horizontal displacement and settlement of the foundation under horizontal dynamic load. The displacement meter adopted a resistance strain displacement sensor with an effective stroke of 200 mm.

In this study, the strain of tower, foundation ring, bottom plate, and bolts connecting foundation ring and tower were mainly considered to measure the dynamic response characteristics of wind turbine extended foundation. Firstly, in order to test the strain of the tower, a test section was arranged at 5cm from the bottom of the tower. The strain gauge is fixed on the outer cylinder wall, and a strain gauge was arranged every 45° on the test section. Secondly, a monitoring section was arranged in the middle of the foundation ring, and a strain gauge was installed every 45° in the section. Thirdly, three measuring lines were arranged on the base plate along the direction of parallel wind direction, one of which passes through the center of the foundation circle, and the direction of each measuring line strain gauge on the base plate is parallel to the wind direction. Fourth, due to the nut connecting the foundation ring and tower surface roughness, bolt surface contact with other parts, strain gauge can not be directly adhered, so the strain gauge bonded to the steel plate for indirect measurement. However, considering that the steel plate is located outside the bolt, the deformation of the steel plate and the bolt is not linear when the bolt is compressed. In this study, strain gauges are attached to the steel plate on the windward side to test the tensile strain of the bolt.

The earth pressure box is arranged on the contact surface between the foundation and the foundation soil to test the variation law of the foundation soil pressure of the wind turbine foundation under random wind load. The range of the earth pressure box used in this experiment is 0~300 MPa. Considering that the wind turbine load is mainly divided into upwind and leeward side, and in the direction perpendicular to the wind load, the model has geometric symmetry, and load is also symmetrical distribution. Therefore, taking the direction of wind load as the center line, the 1/2 foundation on the side of vertical wind load is selected to arrange the earth pressure box, in which the earth pressure box is 5 cm away from the edge of the foundation and the interval is 45°. Next, a soil pressure box is placed in the center of the base, a total of 6 soil pressure boxes, and the specific arrangement is shown in Figure 5.

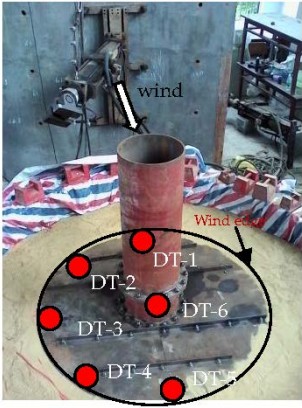

**Figure 5.** Earth pressure box arrangement.

During the test, all the test parts were cleared after the vertical force and eccentric force was applied stably, namely before the horizontal dynamic load is applied. Only displacement, settlement, strain, and pressure caused by horizontal dynamic loads were collected.

### 3. Analysis of Test Results

#### *3.1. Strain of the Connecting Bolt*

Since the surface shape and position of the bolt do not meet the conditions of use of the strain gauge, the strain of the steel plate is approximated by connecting the steel plate to the windward side of the joint to measure the bolt strain. The test results are shown in Figure 6. The axial deformation of the connecting bolt is mainly tensile strain, which fluctuates randomly with time. The maximum strain shown in the figure is 28.3 $\mu\varepsilon$, and the corresponding axial stress is 5.94 MPa according to the calculation formula of elastic deformation. From a numerical point of view, the axial stress caused by horizontal random wind load is small, and this small random cyclic stress may cause fatigue damage of steel, which cannot be completely ignored.

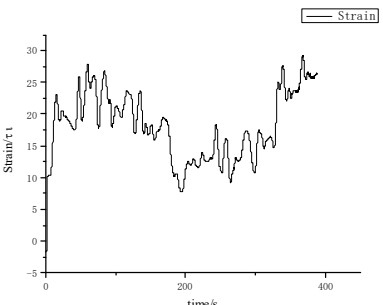

**Figure 6.** Windward side bolt vertical axial curve.

#### *3.2. Vertical Strain in the Middle of the Base Ring*

The windward and leeward strains of the middle section of the foundation ring are shown in Figures 7 and 8. The maximum tensile strain caused by horizontal dynamic load is located on the windward side, and its value is 17.2 $\mu\varepsilon$. The maximum compressive strain caused by horizontal dynamic load is located on the leeward side, and the maximum absolute value is 10.8 $\mu\varepsilon$. According to the formula $\sigma = E\varepsilon$, the maximum tensile stress is 3.66 MPa, the maximum compressive stress is 2.298 MPa, and the stress value is much smaller than the steel yield strength 349.2 MPa. Under the action of random wind load, the windward test of the foundation is mainly subjected to tensile force, while the leeward side is mainly subjected to pressure. In order to ensure the service life of the foundation ring under random wind load, it is necessary to carry out targeted strengthening design.

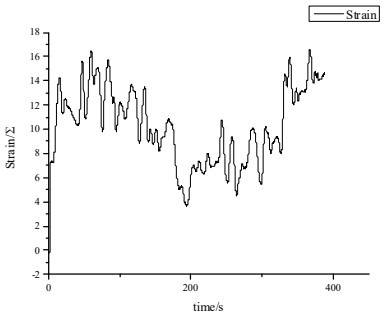

**Figure 7.** Vertical strain curve on the windward side of the base ring.

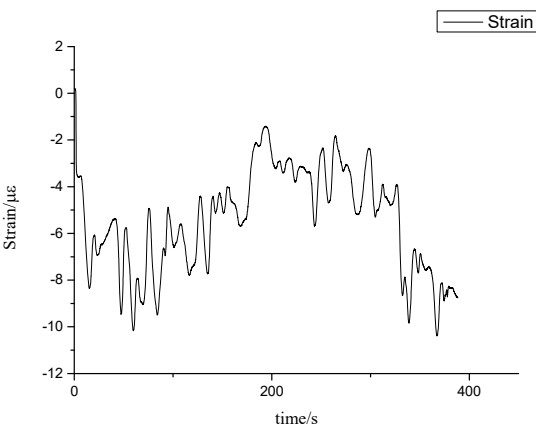

**Figure 8.** The vertical strain on the leeward side of the foundation ring.

### 3.3. Foundation Displacement and Settlement

Horizontal displacement time curve and settlement time curve of the model test are shown in Figures 9 and 10. In the experiment, the horizontal displacement of the model does not exceed 0.50 mm, and the vertical displacement value is less than 0.14 mm. The displacement value of the model experiment is scaled up, and the deformation value is less than the 100 mm allowable range required by the specification.

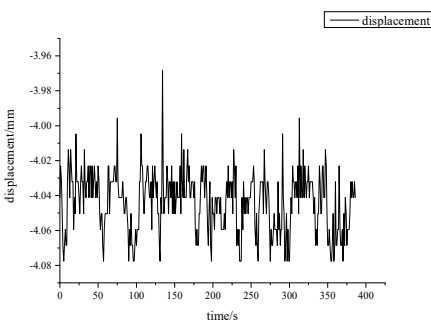

**Figure 9.** Horizontal displacement–time curve of foundation.

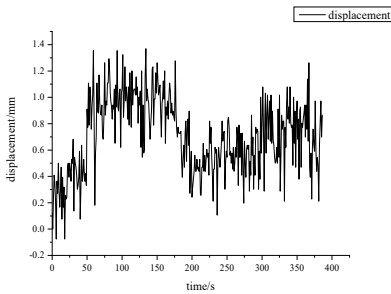

**Figure 10.** Vertical displacement–time curve of foundation.

### 3.4. Strain of the Base Plate

The strain gauges were arranged parallel to the horizontal load direction and along the center line of the bottom of the foundation. The numbers are JY-1 #, JY-2 #, JY-3 #, and JY-4 #. The strain of the bottom plate on the windward side and the leeward side were analyzed. The specific change of the maximum base pressure on the line is shown in Figure 11, where JY-1 # is located on the leeward side and JY-4 # is located on the windward side. It is found that the change value of the bottom plate strain on the windward side is positive, and its tensile strain value is 6.4 με, which is much smaller than the ultimate tensile strain of concrete 100 με; on the leeward side, the strain value of the bottom plate is negative, and the maximum compression value is −10.6 με, which is much smaller than the ultimate

compression strain of concrete 3300 με. It can be found that although the change of strain at the bottom of random foundation does not exceed the limit change value of concrete, its deformation is greatly affected by the direction of wind load.

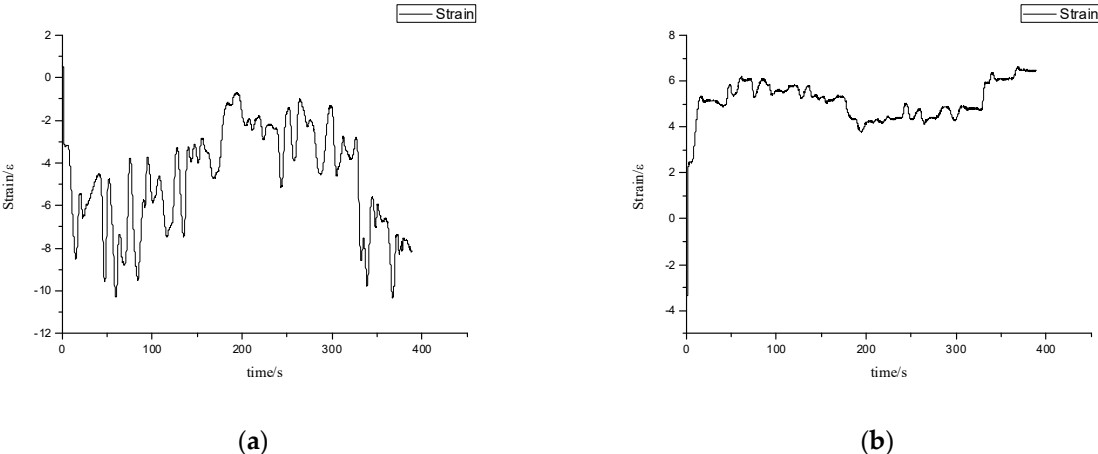

(**a**)                                    (**b**)

**Figure 11.** Base plate maximum strain–position curve. (**a**) JY−1 #; (**b**) JY−4 #.

### 3.5. Foundation Pressure

The base pressure variations on the windward and leeward sides are shown in Figure 12. It can be seen from the figure that the change of foundation pressure caused by horizontal load is quite different. The base pressure generated on the windward side is small, and some data are negative. The horizontal load makes the foundation pressure smaller, while on the leeward side, the base pressure is positive and greater than the base pressure caused on the windward side. The substrate load in the actual operation process is also unevenly distributed, but the specification assumes that the foundation pressure is evenly distributed, which has a certain impact on the design calculation.

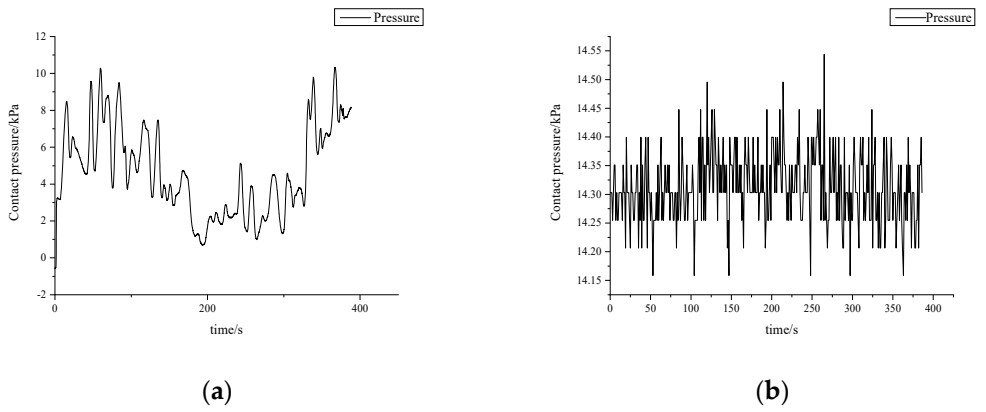

(**a**)                                    (**b**)

**Figure 12.** Foundation bottom pressure curve. (**a**) Windward side substrate pressure curve; (**b**) Basal pressure curve on the leeward side.

## 4. Numerical Simulation

To study the influence of the uneven distribution of substrate pressure on wind power foundation damage, a numerical model was established to study the variation of foundation soil stress and foundation uneven settlement under horizontal wind load.

### 4.1. Establishment of the Numerical Model

According to the analysis in the previous section, the foundation calculation model of the wind turbine meets both the load symmetry condition and the geometric symmetry

condition. Therefore, to simplify the calculation, 1/2 of the geometric model could be numerically modeled. The numerical model established is shown in Figure 13.

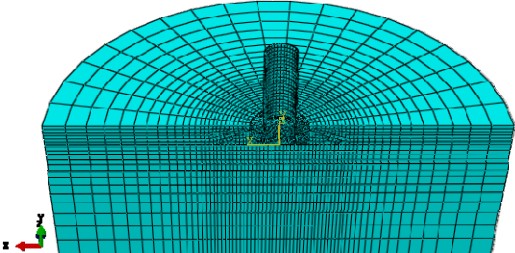

**Figure 13.** Wind turbine ground foundation finite element model.

The infinite element method was used to solve the dynamic boundary problem, that is, the horizontally outward infinite unit CIN3D8 was used on the side of the foundation soil [38]. The established model is show in Figure 14. The vertical force, horizontal resultant force, and horizontal resultant force were applied to the top of the tower according to the simplified wind load and moment.

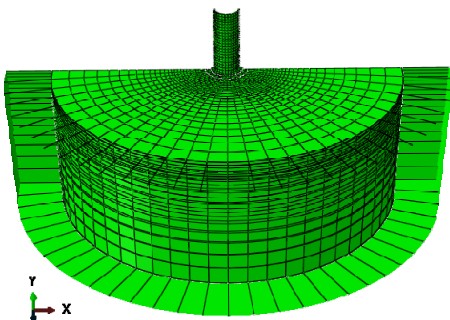

**Figure 14.** Infinite unit artificial boundary.

*4.2. Influence of Wind Load on Foundation Displacement*

Figure 15 is the displacement time curve of the foundation. The horizontal displacement and vertical displacement of the foundation of the numerical simulation and the model test are compared and analyzed. It is found that under the same wind load, the numerical simulation results are similar to the model test results. The numerical simulation results of the horizontal displacement of the foundation are in good agreement with the results of the model test. In the vertical displacement, the model test results at the beginning are larger than the numerical simulation results, but the later values are gradually close.

Through comparative analysis of the model experimental data and numerical calculation results of the base pressure measurement point DT-1 # (as shown in Figure 16a), it is found that the settlement of each point of the base is related to the base pressure to a certain extent, that is, the settlement of the foundation at the location with large base pressure is large, and the settlement at the location with small base pressure is small. Through correlation analysis, the correlation coefficient between them is about 0.8. In terms of numerical value, the numerical simulation results are very close to the experimental data, and their variation trends are the same. This shows that the numerical simulation calculation parameters are selected reasonably, which can better reflect the dynamic response information of the foundation, and can reasonably reflect the deformation characteristics of the wind turbine foundation. In Figure 16b, the vertical displacements of several earth pressure measuring points are compared and analyzed. It can be found that the model test is consistent with the numerical calculation in the change trend, in which the vertical displacement change of DT-3 # is the smallest, while the absolute value of DT-5 # is the

largest. According to the layout of earth pressure measuring points in Figure 16, it can be inferred that the leeward side of the foundation has a large bending settlement, and the windward side has a relatively upward lift.

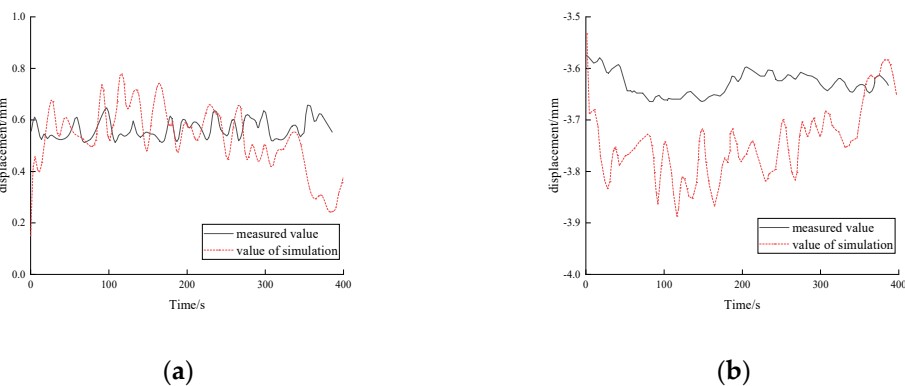

**Figure 15.** Displacement time curve. (**a**) Horizontal displacement–time curve of foundation; (**b**) Vertical displacement–time curve of foundation.

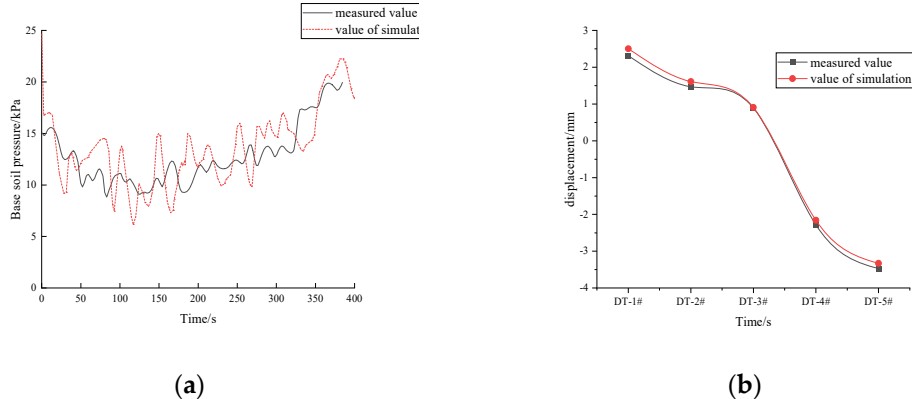

**Figure 16.** (**a**) DT−1 # soil pressure time curve; (**b**) Relative vertical displacement changes curve of each earth pressure measuring point.

### 4.3. Failure Mode under Strong Wind Loads

From the above analysis, it can be seen that the strain change of the bottom plate model is small, and the bottom plate can be regarded as steel body, so that the inclination of the tower can be calculated according to the height of the edge of the bottom plate of the tower, and the inclination angle of the tower can be calculated [39]. When the inclination exceeds the standard, the experiment is stopped. In the numerical calculation, the horizontal wind load and its corresponding horizontal moment are gradually increased by 0.1 kN to test the damage of the wind turbine foundation. When the load increases to a certain value, the finite element calculation does not converge, and the final horizontal load is regarded as the ultimate wind load of the wind turbine foundation. Output fan base damage equivalent plastic strain cloud diagram is shown in Figure 17a. It can be seen from the diagram that the foundation of the wind turbine foundation is destroyed, because the soil at the edge of the windward side enters the plastic zone, the foundation soil is partially yielded, the backfill soil on the foundation is uplifted, the foundation on the windward side is significantly uplifted, and the foundation of the wind turbine is overturned. Figure 17b shows the damage of the experimental foundation of the wind turbine model, and the damage can be in good agreement with the numerical simulation results. Through the study of the wind turbine expansion foundation under the action of wind load, under the action of strong wind load, overturning failure is a dynamic response mode of wind turbines in mountainous areas.

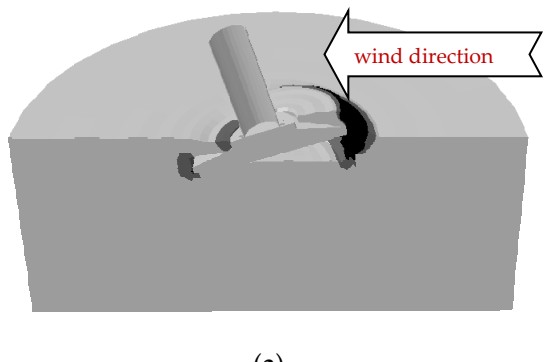
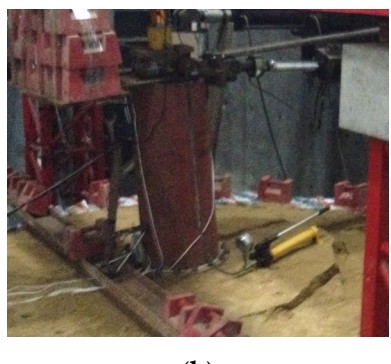

(**a**)                                               (**b**)

**Figure 17.** The failure mode of wind turbine expansion foundation under wind load. (**a**) Equivalent plastic strain nephogram of fan base damage. (**b**) Foundation damage of wind turbine model experiment.

## 5. Conclusions

In this paper, the circular expansion foundation with an installed capacity of 2 MW was taken as the research object, and a scale model test of 1:10 was carried out. The bolt strain, foundation ring strain, foundation base plate strain, foundation displacement, and foundation pressure of the connection between the foundation and the tower are analyzed, and the failure mode of the extended foundation of wind turbine under strong wind load is explored. The main conclusions are as follows:

(1)  Combined with the results of model experiment and numerical calculation, it can be found that the connecting bolts, foundation ring, and foundation bear tensile stress on the windward side of the wind turbine. On the leeward side of the wind turbine, the components of the wind turbine foundation are subjected to compressive stress. Under the action of wind load, the soil of wind turbine foundation bears the influence of dynamic load, and special attention should be paid to the influence of cyclic load on soil strength.

(2)  Under the rated working condition, the bolt strain, foundation ring strain, foundation slab strain, foundation displacement, and foundation pressure fluctuate within a reasonable range, but the relative deformation difference between the windward side and the leeward side is large, and targeted strengthening design is required for the windward side and leeward side.

(3)  The response characteristics of wind turbine foundation under strong wind load are analyzed numerically, and the calculation results are compared with the failure results of scale model experiment. It is shown that the overall overturning failure of foundation is a dynamic response mode of wind turbine foundation, which should be paid attention to in design and construction.

**Author Contributions:** Conceptualization, Z.-W.D. and Z.-J.F.; methodology, Z.-J.F.; software, Z.-J.F.; validation, Y.-M.Z., Z.-W.D.; formal analysis, P.-Y.D.; investigation, Z.-W.D.; resources, Z.-W.D.; data curation, Z.-J.F.; writing—original draft preparation, Z.-W.D.; writing—review and editing, Z.-J.F.; visualization, Y.-M.Z.; supervision, P.-Y.D.; project administration, Z.-W.D.; funding acquisition, Z.-W.D. All authors have read and agreed to the published version of the manuscript.

**Funding:** This work was supported by the Natural Science Foundation of Hunan Province, China (2020JJ4156) and the Postgraduate Research and Innovation Project of Hunan Province (CX20220882).

**Institutional Review Board Statement:** Not applicable.

**Informed Consent Statement:** Not applicable.

**Data Availability Statement:** Not applicable.

**Conflicts of Interest:** The authors declare no conflict of interest.

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
