# Peer review of "Study on Dynamic Response Characteristics of Circular Extended Foundation of Large Wind Turbine Generator"

_sustainability, doi:10.3390/su142114213_

Round 1

Reviewer 1 Report (Previous Reviewer 3)

Dear authors,
I noticed that the suggestions from the previous review have been taken into account, the abstarct, introduction, analysis and references have been completed.
However, I would have liked to see the part of the conclusions more extensive.
In some places, a little improvement of the English language is needed, a suggestion the material needs another detailed reading.

Author Response

Dear Reviewers.

We would like to thank the reviewers for their comments on our manuscript entitled "Dynamic response characterization of the cyclic extension basis of a large wind turbine" (ID: 1991174). These comments are valuable and very helpful in revising and improving our paper and are of great importance to our research. We have carefully studied the comments and made corrections that we hope will be approved. The revised sections are marked in green in the paper. The main corrections and responses to reviewers' comments in the paper are mobile.

Response to reviewer's comments.

Response to comment: I would like to see a broader part of the conclusion.

Response: Based on the reviewer's suggestion and by further summarizing the paper, we have concluded that the windward and leeward sides have different force properties and have included them in the conclusion and abstract.

Response to comment: In some places, some improvements to the English language are needed, and the suggested material needs another detailed reading.

Response: We are very sorry for our writing errors and have re-improved the material part of the article to describe our experimental ideas more clearly.

Special thanks for your kind comments.

We did our best to improve the manuscript and made some changes to it. These changes do not affect the content or the framework of the paper. Here, we do not list these changes, but they are marked in green in the revised paper.

We sincerely thank the reviewers for their enthusiastic work and hope that the corrections will be acknowledged.

Thank you again for your comments and suggestions.

Reviewer 2 Report (Previous Reviewer 2)

Accept in present form

Author Response

We sincerely thank you for your enthusiastic work and encouraging comments.

This manuscript is a resubmission of an earlier submission. The following is a list of the peer review reports and author responses from that submission.

Round 1

Reviewer 1 Report

The authors reported that they performed a research on failure characteristics of circular expansion foundation of large wind turbine. However, after a brief review, I found it behaves badly in writing, format, logic, and novelty. As a scientific paper, I don’t find any expression or quantified conclusion. For the test, very limited information was provided. So, I suggest rejecting it.

Detailed comments:

1.      Line: "a circular expansion foundation model have a ratio of 1:10 to the wind turbine which 2MW installed capacity is established" is not a complete sentence.

2.      Line 19: "he soil"?

3.      Line 19: "is easily disturbed"?

4.      Invite someone with excellent English writing skills to polish this paper. It's hard to read now.

5.      The Introduction section is a mess. There should be connecting and summarizing sentences, instead of listing them out one by one incoherently.

6.      How can you arrange a subsection titled "Simplification of load" in a section named "Material"?

7.      How was the random wind load in Fig. 2 derived? You mentioned that both the mean wind and the random fluctuation of the periodic variation were considered. Elaborate the process.

8.      In Fig. 3, for each level, there should be more points. In addition, provide the expression of the curve.

9.      Besides Fig. 4, there should be a sketch map to show the boundary conditions.

10.   Fig. 5 is unprofessional. Your lovely shadow and shoes are unnecessary.

11.   How was the strain gauge used in Fig. 6 set? The same for the device collecting the data shown in Fig. 8.

12.   The data in Figs. 6 and 7 are in proportional?

13.   In the title of this paper, it writes "failure characteristics", so this experiment was tested until failure? If so, how about the failure criterion?

14.   How many repeated tests were conducted?

15.   How about the theoretical basis or contribution of this work? I don't find any expression.

16.   A CAD plot of the studied object is needed. Mark the detailed dimensions.

17.   The written pattern of this work is more like a project report. Improve it.

18.   Fig. 14 is unprofessional.

19.   In Subsection 3.3, you determined the failure mode purely by simulation?

20.   How about the legend in Fig. 15?

21.   Line 300: "is almost unaffected" is not a scientific conclusion, we need a quantified answer.

22.   Very few of literatures in the reference list were published within recent 5 years. In addition, cite less thesis or journal paper in Chinese. It is unfriendly to international readers.

Author Response

List of Responses

Dear Reviewers:

Thank you for your letter and for the reviewers’ comments concerning our manuscript entitled “Research on Failure Characteristics of Circular Expansion Foundation of Large Wind Turbine” (ID: 1914298).Those comments are all valuable and very helpful for revising and improving our paper, as well as the important guiding significance to our researches. We have studied comments carefully and have made correction which we hope meet with approval. Revised portion are marked in red in the paper. The main corrections in the paper and the responds to the reviewer’s comments are as flowing:

Responds to the reviewer’s comments:

Reviewer #1:

  1. Response to comment:  Line: "a circular expansion foundation model have a ratio of 1:10 to the wind turbine which 2MW installed capacity is established" is not a complete sentence

Response:We have refined the abstract part and modified the expression in it.

  1. Response to comment:Line 19: "he soil"?

Response:We are very sorry for our incorrect writing, we have refined the summary section and corrected the wording error.

  1. 3. Response to comment: Line 19: "is easily disturbed"?

Response:We are very sorry for our incorrect writing, we have refined the summary section and corrected the wording error.

  1. 4. Response to comment: Invite someone with excellent English writing skills to polish this paper. It's hard to read now.

Response:We are very sorry for the negligence in the expression of the article. which has now been revised.

  1. 5. Response to comment:The Introduction section is a mess. There should be connecting and summarizing sentences, instead of listing them out one by one incoherently.

Response:We have re-written this part according to the Reviewer’s suggestion.

  1. 6. Response to comment: How can you arrange a subsection titled "Simplification of load" in a section named "Material"?

Response:The chapters were rearranged as suggested by the reviewers.

  1. 7. Response to comment:How was the random wind load in Fig. 2 derived? You mentioned that both the mean wind and the random fluctuation of the periodic variation were considered. Elaborate the process.

Response:We are very sorry for the negligence of random wind load description.

  1. 8. Response to comment:In Fig.3, for each level, there should be more points. In addition, provide the expression of the curve.

Response:As suggested by the reviewer to add a surface description formula, we have added a non-uniformity coefficient calculation formula in the figure.

  1. 9. Response to comment:Besides Fig.4, there should be a sketch map to show the boundary conditions.

Response:As suggested by the reviewer to add a sketch of the model experiment, we added layout drawings for the experimental model.

  1. 10. Response to comment:Fig.5 is unprofessional. Your lovely shadow and shoes are unnecessary.

Response:As the reviewer suggested that no elements unrelated to the experiment appeared, we replaced the diagram.

  1. 11. Response to comment: How was the strain gauge used in Fig. 6 set? The same for the device collecting the data shown in Fig. 8.

Response:As recommended by the reviewer for the description of the strain gauge placement, the strain gauge placement is shown in the added Figure 4.

  1. 12. Response to comment:The data in Figs. 6 and 7 are in proportional?

Response:The data of Figure 6 and Figure 7 are not proportional.

  1. 13. Response to comment:In the title of this paper, it writes "failure characteristics", so this experiment was tested until failure? If so, how about the failure criterion?

Response:Yes, the experiment went on until it was destroyed, and the destruction criteria for the experiment are described in detail in Section 3.3 of the revision.

  1. 14. Response to comment:How many repeated tests were conducted?

Response:According to the existing experimental error analysis, we conducted three repeated experiments.

  1. 15. Response to comment:How about the theoretical basis or contribution of this work? I don't find any expression.

Response:Through our research, we have a detailed understanding of the dynamic response characteristics of wind turbine foundation, and a more in-depth understanding of the deformation characteristics of foundation under random wind load.

  1. 16. Response to comment:A CAD plot of the studied object is needed. Mark the detailed dimensions.

Response:According to the reviewer 's suggestion, a detailed model experiment diagram is added in Figure 4.

  1. 17. Response to comment:The written pattern of this work is more like a project report. Improve it.

Response:Thanks to the suggestions of reviewers, we have revised the paper.

  1. 18. Response to comment:Fig. 14 is unprofessional.

Response:In this paper, the infinite element method is used to solve the dynamic boundary problem in the dynamic analysis of the wind turbine foundation, that is, the infinite element CIN3D8 with horizontal direction outward is used on the side of the foundation soil. Figure 14 is a similar description of this idea.

  1. 19. Response to comment:In Subsection 3.3, you determined the failure mode purely by simulation?

Response:The destruction of the model is not only determined by numerical calculation, but also by the inclination of the model experiment, which has been described in the modified 3.3 section.

  1. 20. Response to comment:How about the legend in Fig. 15?

Response:We have made correction according to the Reviewer’s comments.

  1. 21. Response to comment:Line 300: "is almost unaffected" is not a scientific conclusion, we need a quantified answer.

Response:We have made correction according to the Reviewer’s comments.

  1. 22. Response to comment:Very few of literatures in the reference list were published within recent 5 years. In addition, cite less thesis or journal paper in Chinese. It is unfriendly to international readers.

Response:We have made correction according to the Reviewer’s comments.

Special thanks to you for your good comments.

We tried our best to improve the manuscript and made some changes in the manuscript. These changes will not influence the content and framework of the paper. And here we did not list the changes but marked in red in revised paper.

We appreciate for Editors/Reviewers’ warm work earnestly, and hope that the correction will meet with approval.

Once again, thank you very much for your comments and suggestions.

Reviewer 2 Report

As considering the subject and results, the study gives some new information. This paper should be developed for publication in this journal following the comments:

Considering the subject and results, the study provides some information.

1.     Check for few minor typos and punctuation mistakes within the text,

2.     This article English of the paper should be polished carefully.

3.     The difference of the study (originality of the study) from the studies in the literature and the aim of the study should be given in the introduction section with clear sentences.

4.  The introduction can be improved.

https://doi.org/10.12989/anr.2022.12.4.405

https://doi.org/10.1590/1679-78251574

Author Response

List of Responses

Dear Reviewers:

Thank you for your letter and for the reviewers’ comments concerning our manuscript entitled “Research on Failure Characteristics of Circular Expansion Foundation of Large Wind Turbine” (ID: 1914298).Those comments are all valuable and very helpful for revising and improving our paper, as well as the important guiding significance to our researches. We have studied comments carefully and have made correction which we hope meet with approval. Revised portion are marked in red in the paper. The main corrections in the paper and the responds to the reviewer’s comments are as flowing:

Responds to the reviewer’s comments:

  1. Response to comment: Check for few minor typos and punctuation mistakes within the text,

Response:We 're sorry we made some small mistakes and used the wrong punctuation. We have modified these issues.

  1. Response to comment:This article English of the paper should be polished carefully.

Response:We have made correction according to the Reviewer’s comments.

  1. 3. Response to comment: The difference of the study (originality of the study) from the studies in the literature and the aim of the study should be given in the introduction section with clear sentences.

Response:We have made corrections based on the reviewer 's comments and carefully revised the introduction.

  1. 4. Response to comment: The introduction can be improved.

Response:We made corrections according to the opinions of the reviewers and carefully revised the citation part.

Special thanks to you for your good comments.

We tried our best to improve the manuscript and made some changes in the manuscript. These changes will not influence the content and framework of the paper. And here we did not list the changes but marked in red in revised paper.

We appreciate for Editors/Reviewers’ warm work earnestly, and hope that the correction will meet with approval.

Once again, thank you very much for your comments and suggestions.

Reviewer 3 Report

Dear authors,

The paper is quite well done, perhaps it should be extended the part of conclusions.

The paper presents a topic of interest for researchers and practitioners. However, several improvements are needed.
1. The abstract should present the need for research, the methodology, the main results obtained and the future directions of research.
2. The introduction should be supplemented with other research conducted in this field and with other developed models developed. In this way the research will show what is the gap it fills and what are the elements of originality.
3. To emphasize the need for this study.
4. To highlight the gaps filled by the present study.
6. The conclusions section should be completed with a review of the study

7. On the analysis side I would suggest that the results be in the form of charts for better visualization and understanding.

Author Response

Dear Reviewers:

Thank you for your letter and for the reviewers’ comments concerning our manuscript entitled “Research on Failure Characteristics of Circular Expansion Foundation of Large Wind Turbine” (ID: 1914298).Those comments are all valuable and very helpful for revising and improving our paper, as well as the important guiding significance to our researches. We have studied comments carefully and have made correction which we hope meet with approval. Revised portion are marked in red in the paper. The main corrections in the paper and the responds to the reviewer’s comments are as flowing:

Responds to the reviewer’s comments:

  1. Response to comment:The abstract should present the need for research, the methodology, the main results obtained and the future directions of research.

Response:We made corrections according to the comments of the reviewers, and carefully revised the abstract.

  1. Response to comment:The introduction should be supplemented with other research conducted in this field and with other developed models developed. In this way the research will show what is the gap it fills and what are the elements of originality.

Response:We have made corrections based on the reviewer 's comments and have carefully revised the introduction.

  1. 3. Response to comment:To emphasize the need for this study.

Response:We made corrections according to the comments of the reviewers, and explained the purpose and significance of our work in detail..

  1. 4. Response to comment:To highlight the gaps filled by the present study.

Response:We made corrections according to the comments of the reviewers, and described the significance of the work in detail..

  1. 5. Response to comment:The conclusions section should be completed with a review of the study.

Response:We have made correction according to the Reviewer’s comments.

  1. 6. Response to comment: On the analysis side I would suggest that the results be in the form of charts for better visualization and understanding.

Response:Considering the Reviewer’s suggestion, we have a lot of work on the chart description and detailed description.

Special thanks to you for your good comments.

We tried our best to improve the manuscript and made some changes in the manuscript. These changes will not influence the content and framework of the paper. And here we did not list the changes but marked in red in revised paper.

We appreciate for Editors/Reviewers’ warm work earnestly, and hope that the correction will meet with approval.

Once again, thank you very much for your comments and suggestions.

Round 2

Reviewer 1 Report

Though the work has been refined according to the first version, however, I think it still far from the level of this journal. 

1) The title should be refined, like clarify the "failure characteristics", what are the main points of this item?

2) The main contribution of this work is not clear,  which should be highlighted in introduction;

3) The entire work structure is not well structured, those load data have no means in the current form, which should be quantified;

4) More discussions on the results in section 3 are needed, and any evidences from the field?

Author Response

Dear Reviewers:

Thank you for your letter and for the reviewers’ comments concerning our manuscript entitled “Study on dynamic response characteristics of circular extended foundation of large wind turbine generator” (ID: 1914298).Those comments are all valuable and very helpful for revising and improving our paper, as well as the important guiding significance to our researches. We have studied comments carefully and have made correction which we hope meet with approval. Revised portion are marked in blue in the paper. The main corrections in the paper and the responds to the reviewer’s comments are as flowing:

Responds to the reviewer’s comments:

  1. Response to comment:The title should be refined, like clarify the "failure characteristics", what are the main points of this item?

Response:It is really true as Reviewer suggested that we have redrafted the title of the article.

  1. Response to comment:The main contribution of this work is not clear, which should be highlighted in introduction;

Response:We have rewritten the contribution of this article in the introduction based on the reviewer 's suggestion.

  1. 3. Response to comment:The entire work structure is not well structured, those load data have no means in the current form, which should be quantified;

Response:We have made correction according to the Reviewer’s comments.

  1. 4. Response to comment:More discussions on the results in section 3 are needed, and any evidences from the field?

Response:We have made correction according to the Reviewer’s comments.

Special thanks to you for your good comments.

We tried our best to improve the manuscript and made some changes in the manuscript. These changes will not influence the content and framework of the paper. And here we did not list the changes but marked in blue in revised paper.

We appreciate for Editors/Reviewers’ warm work earnestly, and hope that the correction will meet with approval.

Once again, thank you very much for your comments and suggestions.

Reviewer 2 Report

Dear Sir

This paper is acceptable.

All corrections have been made.

Best regards

Reviewer 3 Report

Dear authors,

I noticed that the suggestions from my review were taken into account.
Perhaps some of the steps would have been better explained in more detail, but the paper can be published in this form.

Best regards,